# Comparison of diagnostic performance between convolutional neural networks and human endoscopists for diagnosis of colorectal polyp: A systematic review and meta-analysis

**Yixin Xu[1], Wei Ding[1], Yibo Wang[1], Yulin Tan[1], Cheng Xi[1], Nianyuan Ye[1], Dapeng Wu[2], Xuezhong Xu**[1] *

**1** Department of General Surgery, Changzhou Wujin People's Hospital Affiliated to Jiangsu University, The Wujin Clinical College of Xuzhou Medical University, Changzhou, Jiangsu Province, China, **2** Department of Endoscopy, Jiangsu Provincial Hospital of Traditional Chinese Medicine, Nanjing, Jiangsu, China

* xxz197001@sina.com

**Data Availability Statement:** All relevant data are within the manuscript and its Supporting Information files.

## Abstract

Prospective randomized trials and observational studies have revealed that early detection, classification, and removal of neoplastic colorectal polyp (CP) significantly improve the prevention of colorectal cancer (CRC). The current effectiveness of the diagnostic performance of colonoscopy remains unsatisfactory with unstable accuracy. The convolutional neural networks (CNN) system based on artificial intelligence (AI) technology has demonstrated its potential to help endoscopists in increasing diagnostic accuracy. Nonetheless, several limitations of the CNN system and controversies exist on whether it provides a better diagnostic performance compared to human endoscopists. Therefore, this study sought to address this issue. Online databases (PubMed, Web of Science, Cochrane Library, and EMBASE) were used to search for studies conducted up to April 2020. Besides, the quality assessment of diagnostic accuracy scale-2 (QUADAS-2) was used to evaluate the quality of the enrolled studies. Moreover, publication bias was determined using the Deeks' funnel plot. In total, 13 studies were enrolled for this meta-analysis (ranged between 2016 and 2020). Consequently, the CNN system had a satisfactory diagnostic performance in the field of CP detection (sensitivity: 0.848 [95% CI: 0.692–0.932]; specificity: 0.965 [95% CI: 0.946–0.977]; and AUC: 0.98 [95% CI: 0.96–0.99]) and CP classification (sensitivity: 0.943 [95% CI: 0.927–0.955]; specificity: 0.894 [95% CI: 0.631–0.977]; and AUC: 0.95 [95% CI: 0.93–0.97]). In comparison with human endoscopists, the CNN system was comparable to the expert but significantly better than the non-expert in the field of CP classification (CNN *vs.* expert: RDOR: 1.03, *P* = 0.9654; non-expert *vs.* expert: RDOR: 0.29, *P* = 0.0559; non-expert *vs.* CNN: 0.18, *P* = 0.0342). Therefore, the CNN system exhibited a satisfactory diagnostic performance for CP and could be used as a potential clinical diagnostic tool during colonoscopy.

**Funding:** The author(s) received no specific funding for this work.

## Introduction

Based on 2018 reports, colorectal cancer (CRC) had approximately 1,800,000 new cases and 881,000 deaths, implying 1 in 10 cancer cases and deaths [1]. Approximately 85% of CRCs developed from precancerous polyps through genetic and epigenetic mechanisms with a mean dwell time of at least 10 years [2, 3]. Therefore, early and precise detection of colorectal polyp (CP) has a great significance in the prevention of CRC. Notably, colonoscopy is the most effective and essential method in the early diagnosis and prevention of CRC through detection and removal of the neoplastic lesion before its progression to invasive cancer [4]. Reports indicate that the CRC incidence of individuals taken single negative screening colonoscopy was lower by 72% and CRC mortality by 81% than in the general population [5]. Meanwhile, the removal of colorectal polyps could significantly reduce the risk of CRC [6]. Thus, achieving a better diagnostic accuracy of CP for their prevention and better treatment is critical.

Pathologically, CP can be categorized into inflammatory polyp sessile, hyperplastic polyp, serrated adenoma polyp (SSAP), and adenoma [7]. The risk of developing CRC is different for each classification. For instance, several studies have shown that adenoma, similar to SSAP, has the highest risk of developing and progressing to CRC. In contrast, hyperplastic and inflammatory polyp are hardly to develop to CRC [7, 8]. Therefore, how to accurately classify CP remains vital for both the endoscopists and patients, since precise differentiation of CP minimizes unnecessary endoscopic resection, subsequently decreasing the incidence of surgical complications, medical costs, and labor burden of doctors [9].

Despite colonoscopy being effective in the early diagnosis of CRC, it remains imperfect and has several fundamental limitations. First, it has a relatively- high rate of misdiagnosis [10]. Secondly, a few neoplastic lesions remain difficult to detect, even for expert endoscopists [11]. Additionally, the task is time-consuming for the endoscopists and labor-intensive which can result higher costs, specifically in countries with large populations. Lastly, the diagnostic performance of colonoscopy highly banks on the working experience of endoscopists, which varies among individuals. This implies that the diagnostic accuracy of colonoscopy is unstable.

To resolve these shortcomings, several studies have reported the application of artificial intelligence to improve medical diagnosis. For example, convolutional neural networks (CNN) have recently shown significant potential to assist endoscopists causing increased diagnostic accuracy of CP during colonoscopy [12]. Besides, CNN is a type of the most common network architectures of deep learning (DL) methods based on artificial intelligence (AI) technology. Moreover, additional studies showed that the CNN system could automatically classify CP based on its morphological features. It is significantly helpful in the therapeutic decision-making process during colonoscopy [13–15]. Nevertheless, this technology has not reached maturity. Also, a majority of controversies exist on whether the CNN system provides a better performance than the human endoscopists, and whether it is worthy of popularizing.

Here, we compared the diagnostic performance between the CNN system and human endoscopists in the field of CP detection and classification.

## Materials and methods

### Literature search strategy

A systematic literature search was conducted online for studies that assessed the diagnostic value of the CNN system used in the field of colonoscopy for colorectal polyp detection and classification. PubMed, Web of Science, Cochrane Library, and EMBASE databases (up to April 30, 2020) were used during the search with the combination of the following terms:

(["artificial intelligence"] OR ["convolutional neural networks"] OR ["deep learning"] OR ["computer-aided"]) AND (["colonoscopy"] OR ["endoscopy"]) AND (["colon"] OR ["rectum"] OR ["colonic"] OR ["rectal"] OR ["colorectal"]) AND (["polyp"] OR ["polyps"]).

All article sections were carefully reviewed. Subsequently, bibliographies of the retrieved articles were screened to identify any potential source of relevant studies.

### Study selection

The inclusion criteria included (1) studies that included patients with CP; (2) colonoscopy was performed to detect or classify colorectal polyps; (3) CNN system was applied to improve the diagnostic performance of colonoscopy; (4) precise diagnostic data were presented in the article; (5) if the colorectal polyps were classified, the final pathology results were provided. On the other hand, the exclusion criteria included (1) the types of articles were abstracts, reviews, letters, comments, and case reports; (2) precise data were unavailable in the article; (3) animal studies and non-English publications.

### Data extraction and quality assessment

A total of 2 independent researchers (Ye and Xi) conducted the data extraction from the included studies. The information of enrolled studies included the first author's name, publication year, country, diseases concerned, training material, testing material, types of diagnostic performance, and diagnostic performance of the CNN system, expert, and non-expert. The diagnostic performance was categorized as true-positive (TP), false-positive (FP), true-negative (TN), and false-negative (FN) and was retrieved from each article. Moreover, if there was any inconsistency between the 2 reviewers (Ye and Xi), a discussion was conducted including a third investigator to resolve the problem.

Renner *et al*. [15], provided data of "standard-confidence predictions" and "high-confidence predictions". We found that including both of them might introduce the potential of duplication of data. After careful consideration, the "standard-confidence predictions" of data were included. Guo *et al*. [16] provided data of per-frame and per-video and data of per-frame was selected. This was because, first, nearly all of the articles enrolled for analysis used colonoscopy images instead of videos. To ensure the consistency of the whole analysis, the data of per-frame was selected; secondly, the authors did not provide enough per-video data for analysis. Additionally, Wang *et al*. [12], used 4 datasets to validate the diagnostic performance of the CNN system. However, precise data were only provided in Dataset A. As a result, this study chose to include the data of Dataset A. Kudo *et al*. [17], provided both white-light (WLI) and narrow-band image (NBI) for each lesion and tested CNN system in different imaging models. Including both of WLI and NBI images might cause duplication of data. As a result, we deleted the data of NBI images. However, Renner *et al*. [15], Kudo *et al*. [17], and Ozawa *et al*. [18] have the data of diminutive CPs. We thought it was not appropriate to add theme to the general analysis for the potential risk of duplication of data. We initially wanted to perform a subgroup analysis for them, but the STATA software could not do any analysis with sample size smaller than 4.

The methodological quality and applicability of the studies included were evaluated using the quality assessment of diagnostic accuracy scale-2 (QUADAS-2) [19].

### Outcomes of interests

First, pooled sensitivity, specificity, and other diagnostic indices were calculated based on the value of TP, FP, TN, and FN, among CNN system, expert, and non-expert. Secondly, the diagnostic odds ratio (DOR) and the area (AUC) under the summary receive operating

characteristic (SROC) curve, which represented overall diagnostic performance, were examined and compared among different groups. Finally, to identify whether the differences in diagnostic performance were statistically significant, the relative diagnostic odds ratio (RDOR) was compared between each of the 2 groups (CNN system vs. expert; CNN system vs. non-expert; expert vs. non-expert).

## Statistical analyses

Statistical analyses were performed to establish the diagnostic efficacy. The sensitivity, specificity, positive likelihood ratio (PLR), negative likelihood ratio (NLR), DOR, the AUC of SROC, and RDOR were pooled with their 95% confidence interval (CI). A diagnostic tool was considered to have a strong diagnostic value, if its PLR was above 5 and NLR was below 0.2 [20]. The heterogeneity among studies was evaluated by Cochran Q and Higgins' $I^2$ statistics [21]. If the value of $I^2$ was more than 50%, and the value of $P$ less than 0.05, indicating statistically significant heterogeneity existed, a random-effect model was selected for pooling the data [22]. Otherwise, a fixed-effect model was utilized.

SROC was estimated based on the Moses-Littenberg method [23]. Based on the AUC of SROC, the overall diagnostic performance was categorized into 4 levels, including reasonable (<0.75), good (0.75–0.92), very good (0.93–0.96), and excellent (≥0.97) [24].

RDOR was compared between each 2 groups to identify statistically significant differences of diagnostic performance and was based on multivariate meta-regression analysis [25, 26]. The Deeks' funnel plot was used to assess the publication bias.

Pooled sensitivity, specificity, accuracy, PLR, NLR, DOR, and AUC of SROC were calculated using Stata version 14.0. QUADAS-2 assessment was performed using Review Manager version 5.3. The result with a $P$-value of less than 0.05 (p<0.05) was considered statistically significant.

## Results

### Search strategy

Following the initial search through the different databases, a total of 189 articles were identified (102 in PubMed, 31 in Web of Science, 12 in Cochrane Library, and 44 in EMBASE). First, 146 duplicate studies were removed and the remaining 43 articles were screened. In total, 15 articles, including non-English publications, reviews, abstracts, and case reports, which did not meet the inclusion criteria were excluded. Subsequently, the articles with imprecise data and irrelevant subjects were excluded after full-text articles were assessed. Eventually, 13 studies were enrolled in this meta-analysis [12–18, 27–32] (Fig 1). PRISMA flow diagram and checklist are shown in S1 and S2 Tables, respectively.

### Cohort characteristics and quality of included studies

Among the enrolled studies, 7 focused on the field of CP detection, while other studies focused on the field of CP classification. Among these, 5 studies conducted in Japan, 4 in China, 1 in Germany, one in the USA, 1 in Norway, and 1 in Canada, respectively. All articles were published in the last four years (Table 1). Meanwhile, all studies included precise data about the diagnostic performance of the CNN system; 4 studies provided precise data about the performance of experts, and 3 studies provided precise data on the performance of non-expert. All the data about the diagnostic performance of human endoscopists are in the field of CP classification. Histological examination results were the golden standard in the studies done about

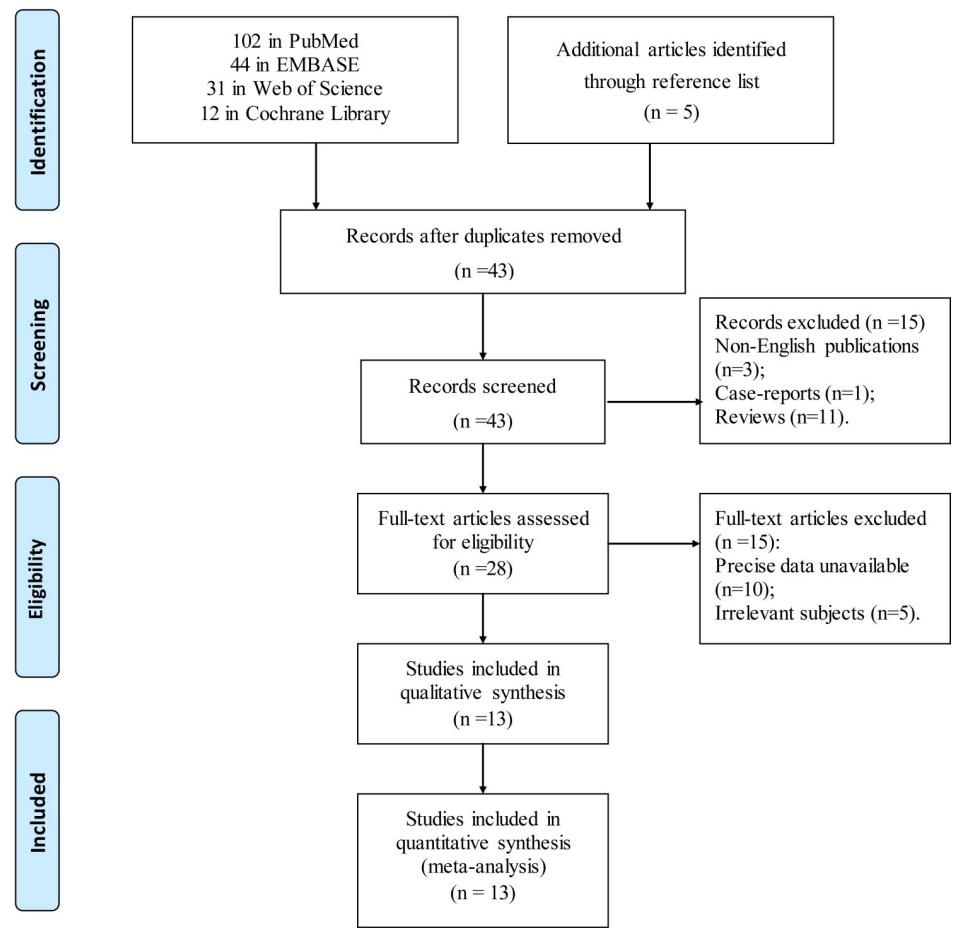

**Fig 1. Flow chart of studies identified, excluded and included.**

CP classification. The diagnostic performance was categorized as TP, FP, FN, and TN (Table 2).

Based on the QUADAS-2 assessment, the quality of all 13 studies included was considered moderate (Fig 2). A total of 11 studies were considered high-quality with low risk in at least 5 of the 7 QUADAS-2 domain. For the patient selection domain, 2 studies introduced bias because case-control design was avoided [18, 27]. Moreover, 3 studies showed a high concern for applicability [13, 17, 18]. Subsequently, for the index test domain, 2 studies had a high risk of bias [13, 17]. Finally, there was only one study that had a high concern regarding reference standard applicability [14].

## Application in the field of colorectal polyp detection diagnostic performance of CNN system

The results of the diagnostic performance of the CNN system are shown in Fig 3. The pooled sensitivity and specificity were 0.848 (95% CI: 0.692–0.932) and 0.965 (95% CI: 0.946–0.977), respectively. The heterogeneity of the sensitivity ($I^2$ = 99.91, $P$ = 0.00) and specificity ($I^2$ = 99.78, $P$ = 0.00) were significant. In addition, the pooled PLR, NLR, and DOR were 24.060 (95% CI: 14.939–38.750), 0.158 (95% CI: 0.073–0.341), and 152.325 (95% CI: 51.654–449.202), respectively. The AUC of SROC of the CNN system was 0.98 (95% CI: 0.96–0.99). The results

**Table 1. Characteristics of the studies included.**

| Author | Year | Country | type of endoscopes | type of CNN system | real-time use of CNN system | type of lesions | type of images | Training material | Testing material | Field focused | Testing objects |
|---|---|---|---|---|---|---|---|---|---|---|---|
| **Lequan** [28] | 2016 | China | images from online database | 3D fully convolutional neural networks | N/A | polyps of any size | N/A | Images | Images | Detection | CNN |
| **Byrne** [27] | 2017 | Canada | 190 series colonoscopes (Olympus) | deep convolutional neural networks | Yes | diminutive polyps | NBI | Videos | Videos | Classification | CNN |
| **Chen** [13] | 2018 | China | CF-H260AZI, PCF-Q260AZI, CF-HQ290AZI (Olympus) | N/A | N/A | diminutive polyps | NBI | Images | Images | Classification | CNN/Expert/Non-expert |
| **Wang** [12] | 2018 | China | Olympus Evis Lucera CV260 (SL)/CV290 (SL) and Fujinon 4400/4450 HD | N/A | Yes | polyps of any size | N/A | Images | Images | Detection | CNN |
| **Renner** [15] | 2018 | Germany | Olympus Evis Exera III CF–HQ 190 colonoscopes | computer-assisted optical biopsy | N/A | polyps of any size | WLI/NBI | Images | Images | Classification | CNN/Expert |
| **Mori** [14] | 2018 | Japan | CFH290ECI colonoscopes (Olympus) | N/A | Yes | diminutive polyps | NBI and methylene blue staining modes | Images | Images | Classification | CNN/Expert/Non-expert |
| **Shin** [29] | 2018 | Norway | images from online database | N/A | N/A | polyps of any size | N/A | Images | Images | Classification | CNN |
| **Urban** [30] | 2018 | USA | PCF-H190 colonoscopes (Olympus) | VGG16,VGG19,and ResNet50 | Yes | polyps of any size | N/A | Images | Images | Detection | CNN |
| **Zhang** [32] | 2018 | China | images from online database | ResYOLO | Yes | polyps of any size | N/A | Images/ | Images | Detection | CNN |
| **Yamada** [31] | 2019 | Japan | images from online database | Faster R-CNN with VGG16 | Yes | polyps of any size | N/A | Images | Images | Detection | CNN |
| **Kudo** [17] | 2019 | Japan | CF-H290ECI (Olympus) | EndoBRAIN | N/A | polyps of any size | WLI/NBI | Images | WLI/NBI images | Detection | CNN/Expert/Non-expert |
| **Guo** [16] | 2020 | Japan | Fujinon 4450 HD | YOLOv3 | Yes | polyps of any size | WLI/NBI | Images | Short/full videos | Detection | CNN |
| **Ozawa** [18] | 2020 | Japan | Evis Lucera and CF Type H260AL/I, PCF Type Q260AI, Q260AZI, H290I, and H290ZI (Olympus) | Single Shot MultiBox Detector | N/A | polyps of any size | WLI/NBI | Images | Images | Classification | CNN |

CNN: convolutional neural networks; NBI: Narrow band imaging; ResYOLO: residual learning modules based on YOLO; YOLO: a CNN system named you only look once; WLI: White light imaging

are as shown in Table 3. Moreover, the PLR and NLR results of the CNN system confirmed that it is an effective method for detecting colorectal polyps.

**Subgroup analysis without the data of short or full videos.** The study of Guo *et al*. [16] included data of videos, and the sample size was large. Considering including it might mislead the general result, we chose to perform a subgroup analysis without it.

**Table 2. A. Diagnostic performance of CNN system, expert, and non-expert in the field of polyp detection. B. Diagnostic performance of CNN system, expert, and non-expert in the field of polyp classification.**

A

| Author | Different grouping standard | CNN system | | | | Expert | | | | Non-expert | | | |
|---|---|---|---|---|---|---|---|---|---|---|---|---|---|
| | | TP | FP | TN | FN | TP | FP | TN | FN | TP | FP | TN | FN |
| **Lequan [28]** | | 3062 | 414 | 9260 | 1251 | | | | | | | | |
| **Wang [12]** | | 6404 | 881 | 20691 | 2345 | | | | | | | | |
| **Urban [30]** | | 7127 | 83 | 1203 | 228 | | | | | | | | |
| **Zhang [32]** | | 3087 | 398 | 13057 | 1226 | | | | | | | | |
| **Yamada [31]** | | 732 | 41 | 4094 | 20 | | | | | | | | |
| **Guo [16]** | Short videos | 2112 | 642 | 21692 | 1608 | | | | | | | | |
| | Full videos | 37938 | 5590 | 78658 | 5672 | | | | | | | | |
| B | | | | | | | | | | | | | |
| | | TP | FP | TN | FN | TP | FP | TN | FN | TP | FP | TN | FN |
| **Byrne [27]** | | 104 | 2 | 19 | 4 | | | | | | | | |
| **Chen [13]** | Group 1 | 181 | 21 | 75 | 7 | 183 | 22 | 74 | 5 | 183 | 29 | 67 | 5 |
| | Group 2 | | | | | 184 | 33 | 63 | 4 | 176 | 33 | 63 | 12 |
| | Group 3 | | | | | | | | | 154 | 22 | 74 | 34 |
| | Group 4 | | | | | | | | | 158 | 11 | 85 | 30 |
| **Renner [15]** | | 48 | 18 | 30 | 4 | 48 | 12 | 36 | 4 | | | | |
| **Mori [14]** | Proximal-rectosigmoid | 167 | 9 | 21 | 12 | 300 | 12 | 48 | 58 | 278 | 20 | 40 | 80 |
| | Rectosigmoid | 95 | 6 | 135 | 5 | 176 | 14 | 268 | 24 | 161 | 30 | 252 | 39 |
| **Kudo [17]** | | 1260 | 0 | 700 | 40 | 603 | 20 | 330 | 20 | 920 | 40 | 460 | 380 |
| **Ozawa [18]** | | 1073 | 175 | 74 | 99 | | | | | | | | |
| **Shin [29]** | | 180 | 13 | 157 | 16 | | | | | | | | |

CNN: convolutional neural networks; FN: false-negative; FP: false-positive; NBI: narrow band imaging; TN: true negative; TP: true positive; WLI: white light imaging.

The resutl showed that the pooled sensitivity and specificity were 0.878 (95%CI: 0.702–0.956) and 0.968 (95%CI: 0.945–0.981). Meanwhile, the PLR, NLR, DOR, and AUC of SROC were 27.314 (95%CI: 14.985–49.788), 0.126 (95%CI: 0.047–0.338), 216.250 (95%CI: 53.307–877.255), and 0.98 (95%CI: 0.97–0.99), respectively. The result was shown in the S3 Table.

## Application in the field of colorectal polyp classification

**Diagnostic performance of CNN system.** First, the pooled sensitivity and specificity were 0.943 (95%CI: 0.927–0.955) and 0.894 (95%CI: 0.631–0.977) (Fig 4). The heterogeneity of sensitivity ($I^2 = 94.77$, $P = 0.00$) and specificity ($I^2 = 98.91$, $P = 0.00$) were significant. Meanwhile, the PLR, NLR, DOR, and AUC of SROC were 8.911 (95%CI: 2.110–37.622), 0.064 (95%CI: 0.043–0.094), 139.052 (95%CI: 22.978–841.481), and 0.95 (95%CI: 0.93–0.97), respectively.

**Diagnostic performance of expert and non-expert.** For the diagnostic performance of expert in the field of classification of colorectal polyps, the pooled sensitivity, specificity, PLR, NLR, DOR and AUC of SROC of expert were 0.944 (95%CI: 0.892–0.972), 0.848 (95%CI: 0.732–0.919), 6.198 (95%CI: 3.416–11.247), 0.066 (95%CI: 0.034–0.127), 94.383 (95%CI: 39.547–225.251), and 0.96 (95%CI: 0.94–0.98), respectively. The heterogeneity of sensitivity ($I^2 = 93.68$, $P = 0.00$) and specificity ($I^2 = 94.03$, $P = 0.00$) were significant.

Besides, the pooled sensitivity, specificity, PLR, NLR, DOR and AUC of SROC of non-expert were 0.859 (95%CI: 0.769–0.918), 0.811 (95%CI: 0.718–0.878), 4.544 (95%CI: 3.122–6.614), 0.174 (95%CI: 0.109–0.277), 26.191 (95%CI: 15.870–43.225), and 0.90 (95%CI: 0.87–

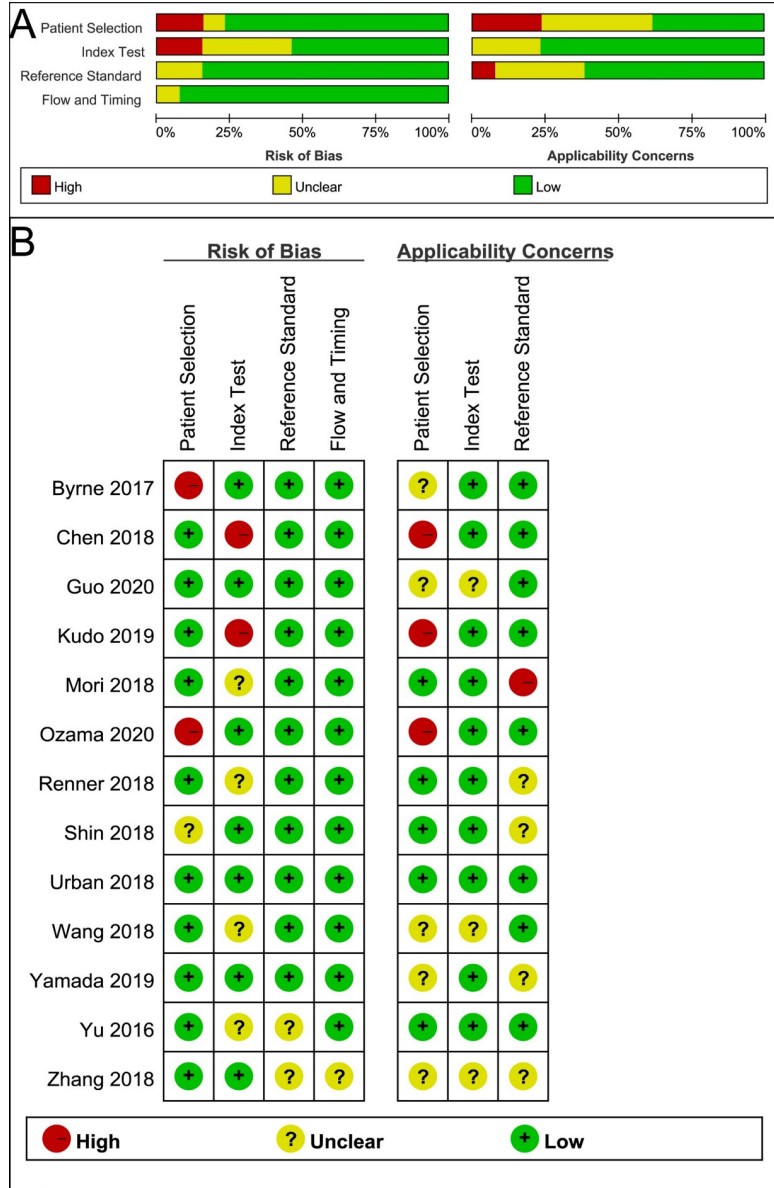

**Fig 2. Methodological quality of the included 13 studies using assessment tool of QUADAS-2.** (A) Grouped bar charts of risk of bias (left) and concerns for applicability (right). (B) Quality assessment for each individual study. QUADAS-2 = Quality Assessment of Diagnostic Accuracy Studies-2.

0.93), respectively. The heterogeneity of sensitivity ($I^2$ = 91.38, $P$ = 0.00) and specificity ($I^2$ = 88.75, $P$ = 0.00) were significant.

All data is summarized in Table 3.

**The comparison of diagnostic performance among CNN system, expert, and non-expert.** For CP classification, the AUC of SROC of CNN, expert, and non-expert was 0.95 (95%CI: 0.93–0.97), 0.96 (95%CI: 0.94–0.98), and 0.90 (95%CI: 0.87–0.93), respectively (Fig 5). By comparing them in pairs acording to RDOR, we found the diagnostic performance of CNN is comparable to that of the expert, but significantly better than that of the non-expert. (Table 4).

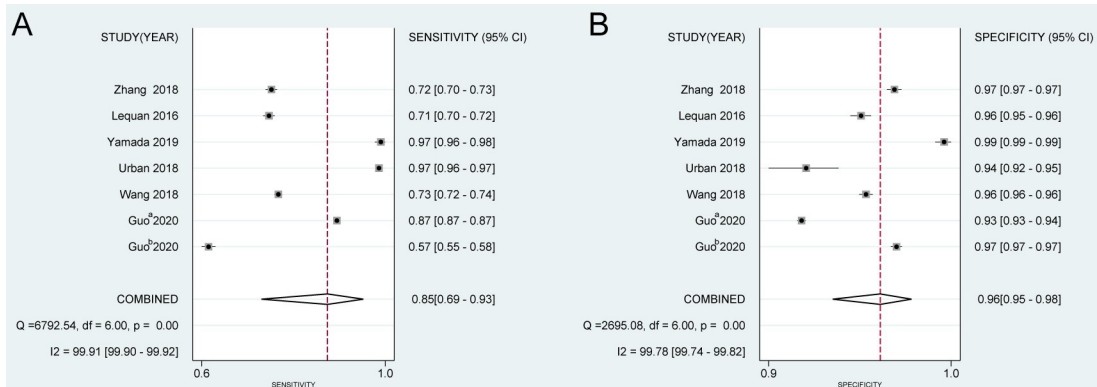

**Fig 3. The pooled diagnostic accuracy index of CNN system in the field of CP detection.** (A) Sensitivity, (B) specificity. a: full videos; b:short videos. CNN: convolutional neural networks; CP: colorectal polyps. NBI: narrow-blue images; WLI: white-light images.

## Publication bias and identification of sources of heterogeneity

According to Deeks' funnel plot asymmetry, no publication bias was reported in pooled results of the CNN system. For CP detection, the result was $P > |t| = 0.430$. At the same time, for CP classification, the result was $P > |t| = 0.196$. They are as shown in Fig 6A and 6B. Since notable heterogeneity was observed in the pooled analysis of the CNN system in the field of CP detection and classification, meta-regression was conducted to identify the source of heterogeneity. Nonetheless, no potential sources of heterogeneity were identified.

## Discussion

This work systematically reviewed the current status of the CNN system applied in the field of CP detection and classification. Moreover, we conducted a quantitative comparison of the diagnostic value between the CNN system and human endoscopists. Our major finding was that the diagnostic performance of the CNN system was comparable to that of the expert in the field of CP classification. In contrast, the performance of the CNN system was significantly superior to that of the non-expert.

The American Society of Gastrointestinal Endoscopy published the Preservation and Incorporation of Valuable Endoscopic Innovations (PIVI) statement in 2015 to address the resect and discard strategy [33]. This approach set the threshold of a diagnose-and-leave strategy for small colorectal polyps at NPV≥90%. At the same time, the threshold of a resect-and-discard strategy was above 90% of the agreement with histopathology for post-polypectomy

**Table 3. Diagnostic performance of CNN system, expert, and non-expert in the field of colorectal polyp classification.**

| Object | Sensitivity (95% CI) | Specificity (95% CI) | PLR (95% CI) | NLR (95% CI) | DOR (95% CI) | SROC (95% CI) |
|---|---|---|---|---|---|---|
| CNN | 0.943 [0.927–0.955] | 0.894 [0.631–0.977] | 8.911 [2.110–37.622] | 0.064 [0.043–0.094] | 139.052 [22.978–841.481] | 0.95 [0.93–0.97] |
| Expert | 0.944 [0.892–0.972] | 0.848 [0.732–0.919] | 6.198 [3.416–11.247] | 0.066 [0.0.34–0.127] | 94.383 [39.547–225.251] | 0.96 [0.94–0.98] |
| Non-expert | 0.859 [0.769–0.918] | 0.811 [0.718–0.878] | 4.544 [3.122–6.614] | 0.174 [0.109–0.277] | 26.191 [15.870–43.225] | 0.90 [0.87–0.93] |

CNN: convolutional neural networks; DOR: diagnostic odds ratio; NLR: negative likelihood ratio; PLR: positive likelihood ratio; SROC: summary receive operating characteristic.

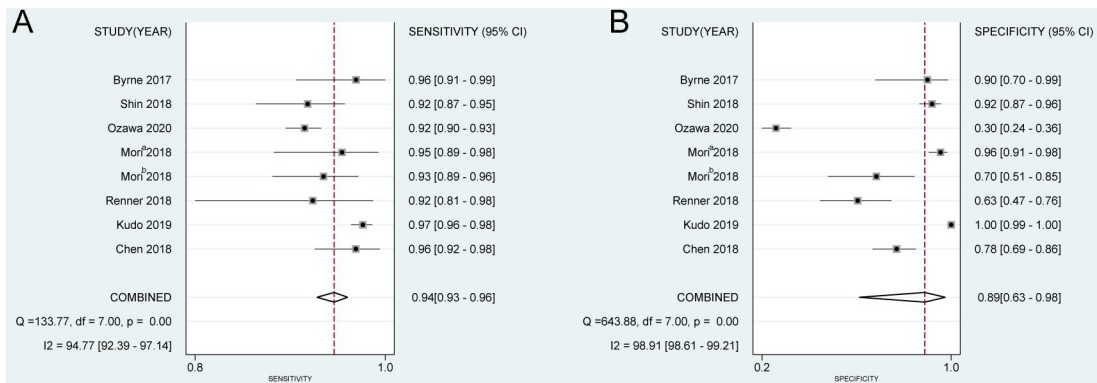

**Fig 4. The pooled diagnostic accuracy index of CNN system in the field of CP classification.** (A) Sensitivity, (B) specificity. a: rectosigmoid; b: proximal-rectosigmoid. CNN: convolutional neural networks; CP: colorectal polyps.

surveillance intervals [34]. These set standards were significantly high and hard to achieve, even for experienced endoscopists. Besides, the task of endoscopists was time-consuming as well as labor-intensive. A few studies have shown that endoscopic detections and predictions triggered a rather low diagnostic accuracy rate, particularly in the case of non-expert use [35, 36]. Hence, this calls for the application and use of technological support. This is because evidence has ascertained that computer-aided diagnosis of endoscopic images using AI has the potential to surpass the diagnostic accuracy of trained specialists. Also, AI might also provide more accurate results without interobserver differences, especially between experts and non-experts.

A considerable number of studies have currently focused on the development of the CNN system that assisted human endoscopists. In the field of colonoscopy, its function is primarily divided into 2 categories, i.e.: detection and classification. For CP detection, we found that the PLR, NLR, and AUC of the CNN system was 8.911 [95%CI: 2.110–37.622], 0.064 [95%CI: 0.043–0.094], and 0.95 [95%CI: 0.93–0.97], respectively. These results suggested that CNN was a good diagnostic tool for CP detection. Guo *et al.* [16] provided the data of videos with large sample size. Considering including them would add potential risk to mislead the general result of CNN, we subsequently performed a subgroup analysis without them. The result just slightly changed which meant it was stable with or without the data of Guo *et al.* [16].

Unfortunately, we didn't find data on human endoscopists in the field of CP dectection. However, some studies demonstrated that non-expert endoscopists could produce a better diagnostic performance during endoscopy after the AI training course [37, 38]. Hence, the AI

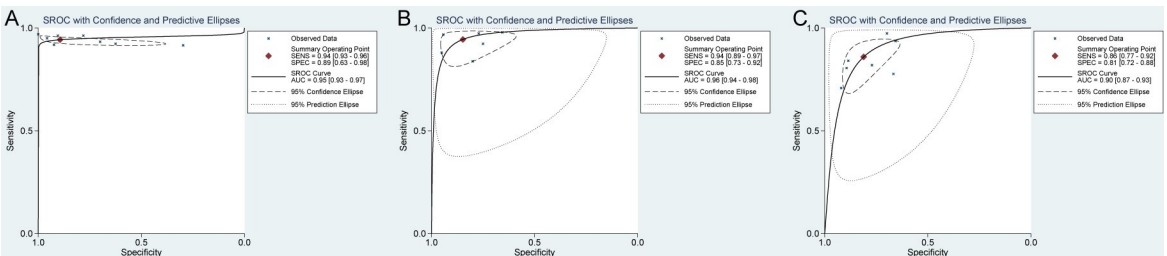

**Fig 5.** Summary receiver operation characteristic (SROC) curve of diagnostic performance of CNN system (A), expert (B), and non-expert (C) for CP classification.

**Table 4. Comparison of diagnostic performance among CNN, expert, and non-expert in the field of colorectal polyp classification.**

| Object | Coefficient | Stand error | RDOR | 95% CI | P |
|---|---|---|---|---|---|
| CNN vs. Expert | 0.033 | 0.7425 | 1.03 | 0.20–5.30 | 0.9654 |
| CNN vs. Non-expert | -1.696 | 0.7099 | 0.18 | 0.04–0.86 | 0.0342 |
| Expert vs. Non-expert | -1.250 | 0.5784 | 0.29 | 0.08–1.04 | 0.0559 |

CNN: convolutional neural networks; RDOR: relative diagnostic odds ratio.

technologies harbor the application potential as a clinical ancillary diagnostic tool and also as an endoscopist training method.

Furthermore, it would be highly beneficial if endoscopic observation can distinguish neoplastic CP from hyperplastic CP. This is because the removal of lesions without malignant potential is expensive and causes high post-procedure complications [39]. Thus, a precise classification of CP significantly improves the cost-effectiveness of colonoscopy. Nonetheless, the task of precisely classifying the different types of CP remains rather difficult. For instance, lesions with indistinct borders, flat and depressed features in conventional adenomas are challenging to distinguish from surrounding normal mucosa. This scenario is specifically prevalent when the bowel preparation is inadequate or the mucosa is capped by mucus or intestinal residue [11]. Kuiper *et al.* revealed that the sensitivity/specificity of classification of diminutive CP was only 77.0%/78.8%, which was far from satisfactory [40]. In this study, we found that the sensitivity/specificity of a non-expert in the field of CP classification was 85.9%/81.1%. As such, the benefits of optical CP classification might remain limited to experts. However, not every endoscopist is an expert. Therefore, the emergence of AI technology has significantly resolved this limitation. Further, we discovered that the diagnostic performance of the CNN system was significantly better than that of the non-expert. However, due to the complexity of classification technology, the DOR of the CNN system applied in the field of CP classification (139.052 [95%CI: 22.978–449.202]) was weaker compared to that in the field of CP detection (152.325 [95%CI: 51.654–449.202]). Alaso, a similar CNN-DL system was used for the diagnosis and classification of proximal gastric precancerous conditions, including chronic atrophic gastritis, intestinal metaplasia, and dysplasia [41]. This system achieved a sensitivity of 93.5%, and an accuracy of 98.3%, which were much better than both the less and more experienced endoscopists.

However, the CNN system was, in essence, a type of algorithm, which could not make logical decisions like humans. It can be used as a training or auxiliary tool to enhance the

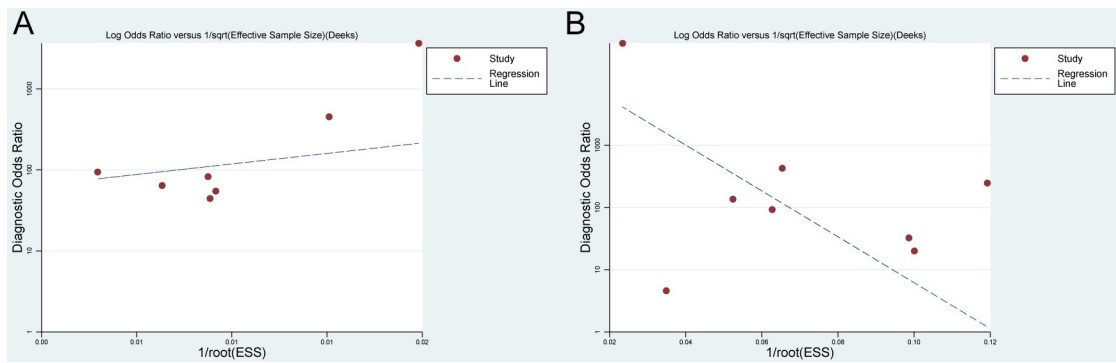

**Fig 6. Deeks' funnel plot for publication bias.** (A) CNN system for CP detection, (B) CNN system for CP classification.

performance of endoscopists, but cannot entirely replace human endoscopists. Besides, CNN technology has several limitations.

First, most of the images and videos extracted for CNN system training are highly qualified, which usually triggers selection bias. These systems are frequently unable to distinguish lesions from low-quality materials. Also, their diagnostic performance is excellent in the training set but weak in the clinical practice.

Secondly, identification of images and videos of rare lesions including subtle flat colonic lesions and morphology types is challenging. They are insufficient in either hospitals' independent or online databases, hence, inadequate training of the CNN system. This further triggers high misdiagnosis rates of infrequent diseases.

Thirdly, most studies included in the present review trained their CNN systems with stationary images or image frames extracted from colonoscopic videos which might hinder the ability of real-time implementation of the CNN system. Moreover, due to the lack of calculating power of computer processors and the complexity of technical processes, the latency of the decision-making process in most systems was unsatisfactory, subsequently disturbing the endoscopist during colonoscopy. Therefore, the ability to work in real-time during endoscopy should be incorporated.

Finally, the CNN system and other artificial intelligence are typically types of algorithm which make decision based on past information. This means it cannot make logical or X crossed decisions. Notably, AI excels when data and training are abundant and exhaustive. However, its performance becomes poorer when it faces previously unseen features and objects since it struggle to extrapolate knowledge gathered from the past to the new environment [42]. In this scenario, humans appear to perform better than AI [43].

With the rapid advancements of AI technology, an ideal CNN system will be developed to overcome these limitations. It might precisely distinguish different lesions from normal surrounding mucosa, including those rare lesions. Meanwhile, it might assist endoscopists simultaneously during endoscopy with almost undetectable latency. Even more, it might provide the type, location, size, depth, and other relevant information of lesions.

In the present study, there are some limitations that should be acknowledged here. First, studies on this field are limited since the application of the CNN system in the field of endoscopy has not matured. Secondly, the sample size of the comparison between the CNN system and human endoscopists was small, which might cause selection bias. Thirdly, although there was no publication bias, since letters, reviews, as well as articles not published in English were excluded, selective reporting bias might still exist. Fourthly, although meta-regression analysis was performed to identify the potential sources of heterogeneity, due to the limitation of the sample size and the variables collected from studies included, the exploration of heterogeneity might remain inadequate. Finally, the majority of studies included were retrospective and used different types of training and testing materials, hence a potential bias.

## Conclusion

In conclusion, our systematic review and meta-analysis suggested that the CNN system achieved comparable diagnostic performance to that of an expert, and better performance compared to that of a non-expert, in the field of CP detection. Additionally, in the field of CP classification, the CNN system demonstrated better diagnostic performance than the human endoscopists regardless of the level of working experience. Despite the limitations of the CNN system, it can be popularized in clinical practice with relative-high diagnostic accuracy, consequently enhancing the diagnostic performance of endoscopists.

## Supporting information

**S1 Table. PRISMA flow diagram.**
(DOC)

**S2 Table. PRISMA checklist.**
(DOC)

**S3 Table. Subgroup analysis without the data of short or full videos in the field of CP detection.**
(DOCX)

## Acknowledgments

The authors thank Dr. Peng Jiang and Dr. Hai-Feng Tang for their critical reading and informative advice during the study process. Meanwhile, the authors thank Freescience for language polishing.

## Author Contributions

**Conceptualization:** Yixin Xu.

**Data curation:** Yixin Xu, Yulin Tan.

**Formal analysis:** Wei Ding, Cheng Xi, Dapeng Wu.

**Methodology:** Wei Ding, Cheng Xi.

**Software:** Nianyuan Ye.

**Writing – original draft:** Yixin Xu, Yibo Wang.

**Writing – review & editing:** Xuezhong Xu.

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
