## [Decision Letter · Decision Letter 0]

22 Sep 2020

PONE-D-20-22694

Comparison of diagnostic performance between convolutional neural networks and human endoscopists for diagnosis of colorectal polyp: a systematic review and meta-analysis

PLOS ONE

Dear Dr. Xu,

Thank you for submitting your manuscript to PLOS ONE. After careful consideration, we feel that it has merit but does not fully meet PLOS ONE’s publication criteria as it currently stands. Therefore, we invite you to submit a revised version of the manuscript that addresses the points raised during the review process.

We look forward to receiving your revised manuscript.

Kind regards,

Ping He, Ph.D.

Academic Editor

PLOS ONE

Journal Requirements:

2. Please include the date(s) on which you accessed the databases or records to obtain the data used in your study.

3. Thank you for quantifying study heterogeneity.

Please provide more detailed reporting on your results, for example, by reporting your Q or I^2 statistics.

4. We note that you state in your manuscript "The design and protocol of the present study were approved by our institutional review board."

Since this is a systematic review and does not include non-public data, this statement is not necessary and may be taken out.

If you would like to keep this statement, please provide the full name of your ethics committee and the ethics approval number.

6. Please include captions for your Supporting Information files at the end of your manuscript, and update any in-text citations to match accordingly. Please see our Supporting Information guidelines for more information: http://journals.plos.org/plosone/s/supporting-information

Additional Editor Comments:

There are serious problems with the manuscript. Please carefully polish and revise your manuscript according to the comments of the reviewers.

Reviewers' comments:

Reviewer's Responses to Questions

**Comments to the Author**

1. Is the manuscript technically sound, and do the data support the conclusions?

Reviewer #1: Yes

Reviewer #2: No

Reviewer #3: Yes

2. Has the statistical analysis been performed appropriately and rigorously? 

Reviewer #1: Yes

Reviewer #2: No

Reviewer #3: Yes

3. Have the authors made all data underlying the findings in their manuscript fully available?

Reviewer #1: Yes

Reviewer #2: Yes

Reviewer #3: Yes

4. Is the manuscript presented in an intelligible fashion and written in standard English?

Reviewer #1: Yes

Reviewer #2: No

Reviewer #3: Yes

5. Review Comments to the Author

Reviewer #1: This is an interesting systematic review article written by Xu et al. This review is well-written and described using appropriate statistical methods. The authors conducted meta-analysis to compare the diagnostic performance of artificial intelligence (AI) for detecting and classifying the colorectal polyp to the human endoscopists. This study must be of interest to the reader and will provide useful knowledge to the reader. I have one comment for the authors.

1. Shouldn't the classification of polyps be divided into normal observation, magnification (80×) and super magnification (520×, microscopic observation) for analysis? Please revisit and consider this point.

Reviewer #2: The meta-analysis presented by Dr Xu and colleagues addresses an interesting topic, that is the performance of convolutional neural networks for detection and classification of colorectal polyps.

The fact that computer-aided colonoscopy and artificial intelligence are hot-topics is demonstrated by the very recent publication of other dedicated systematic reviews and meta-analysis, such as Barua et al, Endoscopy 2020 or Hassan et al, GIE 2020 (detection) and Lui et al, GIE 2020 (detection and classification).

In this meta-analysis, the authors only focus of CNNs, a specific type of artificial intelligence.

Unfortunately, the paper presented by dr. Xu has some important limitations:

- The search strategy seems well designed, and I appreciated the effort to analyze the quality of included studies and the possibility of publication bias. However, at least one recent full-study about CNNs that met the inclusion criteria was not included in the analysis (Urban 2018, Gastroenterology); also some abstracts (such as Misawa 2019 GIE, Matsui 2019 GIE, LUI 2019 GIE) were not included. The choice whether to include or not abstracts can be discussed (even if in a meta-analysis they are usually included), however abstracts were not included in the exclusion criteria.

- Data extraction from the selected studies was not so rigorous and clear.

For example, apparently for reference 13 (Renner et al) authors have included the performances of CNN considering either high-confidence predictions or "standard" predictions (see table 2), with potential duplication of data and overestimation of performance.

Moreover, for reference 25 (Guo et al) the authors considered per-frame sensitivity and specificity of short and long videos; however, in order to provide clinically useful information (similar to polyp or adenoma detection rates) per-video sensitivity and specificity (reported in table 3 of the cited paper for the 100 short videos) should have been used instead. In table 3 of the paper by Guo et al are also reported the diagnostic performances of 2 experts e 2 non-experts physicians that were not included in the metanalysis.

Similarly, for Wang et al (ref 31) the 1633 polyps (and not the total number of polyps images) should have been considered.

Considering these problems in data extraction, the reported pooled diagnostic performances of CNN and human endoscopists are not reliable.

- I think that the use of the Fagan Nomogram to calculate post-test probability considering positive and negative likelihood ratios is not really indicated in this field.

- The differential performance according to dimension of polyps (diminutive and non-diminutive polyps) should have been analysed

- The paper requires a deep linguistic revision (grammatical mistakes, repetitions…), possibly by an English mother-tongue revisor.

Other observations:

- Reference 10 does not refer to detection of colonic lesions, but of early esophageal squamous cell carcinoma/preneoplastic lesions in the esophagus. The sentence in the introduction related to this reference should be revised.

- Both table 1 and 2 should contain the reference numbers of the included papers in order to make the tables easier to read. Moreover, I suggest to list the papers according to the year of publication.

- Table 1 should also include more details about the imaging modalities included in the different studies. For example, the use of dedicated endocytoscopes in the studies referred as 12 and 25 should be reported. Also information about the use of real-time analysis by the CNN systems, and the type of lesions included (only diminutive polyps or polyps of any size) in the different studies should be highlighted in the table.

- In the paper it is stated that 7 of the included studies focused on detection; however in table 1 only 6 papers have detection been reported as field focus; similarly, 5 studies should include experts human endoscopists but only four papers are reported in table 1 and 2.

- The paper by Yu (2016), included in table 1 and 2 is not reported in the references.

Reviewer #3: There are some spelling mistakes:

- line 186, page 15: Results instead of resules

- line 211, page 17, table 2: proximal rectosigmoid instead of poximal rectosigmoid

- line 360, page 25: ordinary physicians instead of phyisicians

There are missing the technical explanations of the different CNN-systems you compared in your analysis: Why are some CNN-systems better for CP detection and others for CP classification? Is it justified to compare different CNN-systems that provide different features? Do they have different characteristics concerning deep learning?

There is also missing the description of the different classification-systems of CP used by different authors in different studies.

6. PLOS authors have the option to publish the peer review history of their article (what does this mean?). If published, this will include your full peer review and any attached files.

Reviewer #1: **Yes: **Naoki Hosoe

Reviewer #2: No

Reviewer #3: No

---

## [Author Response · Author response to Decision Letter 0]

19 Oct 2020

Dear Editor:

 Thank you and the reviewers for your valuable suggestions. We have carefully read through the comments and made proper revisions. Our responses to the reviewer’s questions are listed below. We greatly appreciated your time and efforts to improve our manuscript for publication.

Sincerely,

Yixin Xu

Journal Requirements:

Response:

We have revised the manuscript following the style requirements.

2. Please include the date(s) on which you accessed the databases or records to obtain the data used in your study.

Response: 

We completed the data collection on April 30, 2020. It was mentioned on Page 6, Line 117.

3. Thank you for quantifying study heterogeneity.

Please provide more detailed reporting on your results, for example, by reporting your Q or I^2 statistics.

Response:

In the field of CP detection, we add the value of I2 and P for CNN system on Page 14, Line 248-249; for expert on Page 15-16, Line 274-276; for non-expert on Page 16, Line 282-284.

In the field of CP classification, we add the value of I2 and P for CNN system on Page 16, Line 289-290; for expert on Page 17, Line 309-310; for non-expert on Page 17, Line 314-315.

All these data added were marked in green. 

4. We note that you state in your manuscript "The design and protocol of the present study were approved by our institutional review board."

Since this is a systematic review and does not include non-public data, this statement is not necessary and may be taken out.

If you would like to keep this statement, please provide the full name of your ethics committee and the ethics approval number.

Response:

We have taken out this statement “The design and protocol of the present study were approved by our institutional review board.”, and added “As the data in this study was from previous studies, there was no need to get the ethics committee approval, follow the Declaration of Helsinki, or have patients informed consent form.” to the Ethical approval part. 

Response:

We have added the ORCID to the Editorial Manager. 

6. Please include captions for your Supporting Information files at the end of your manuscript, and update any in-text citations to match accordingly. Please see our Supporting Information guidelines for more information: http://journals.plos.org/plosone/s/supporting-information

Response:

We have added the supporting information at the end of the manuscript and cited on Page 10, Line 206. 

Reviewer 1#

This is an interesting systematic review article written by Xu et al. This review is well-written and described using appropriate statistical methods. The authors conducted meta-analysis to compare the diagnostic performance of artificial intelligence (AI) for detecting and classifying the colorectal polyp to the human endoscopists. This study must be of interest to the reader and will provide useful knowledge to the reader. I have one comment for the authors.

1. Shouldn't the classification of polyps be divided into normal observation, magnification (80×) and super magnification (520×, microscopic observation) for analysis? Please revisit and consider this point.

Response: 

Thanks a lot for your summary and kind remarks. 

To your constructive suggestion, we have carefully reviewed all the references in our meta-analysis. We found that only few studies included in our analysis mentioned the magnification type of their images used to train the CNN system. Only Kudo et al. and Mori et al. used the super magnification (520×). As a result, we did not divide the classification into different groups. Besides, we found that the classification system of CNN applied in colonoscopy mainly focused on determining whether the polyps, especially those diminutive polyps, were hyperplastic or neoplastic. For neoplastic polyps, simple resection may be not sufficient. Endoscopic Mucosal Resection (EMR), Endoscopic submucosal dissection (ESD) or even surgery is the optimal curative option. The process of classification of polyps were finished using the routine colonoscopy images (some systems using narrow-band images). Besides, magnification colonoscopy is currently not commercially available worldwide. 

Reviewer 2#

The meta-analysis presented by Dr Xu and colleagues addresses an interesting topic, that is the performance of convolutional neural networks for detection and classification of colorectal polyps.

The fact that computer-aided colonoscopy and artificial intelligence are hot-topics is demonstrated by the very recent publication of other dedicated systematic reviews and meta-analysis, such as Barua et al, Endoscopy 2020 or Hassan et al, GIE 2020 (detection) and Lui et al, GIE 2020 (detection and classification).

In this meta-analysis, the authors only focus of CNNs, a specific type of artificial intelligence.

Unfortunately, the paper presented by dr. Xu has some important limitations:

Response: 

Thank you for this summary of our paper. 

I started my manuscript on January 10, 2020. Before that, I have not found any similar article through Searching PubMed and other database. I thought this subject about artificial intelligence applied in the field of colonoscopy was interesting and “hot”. I finished my manuscript on June, and submitted it to PLOS ONE on July 22. I have not realized that there were three articles about this subject have already been published. Those limitations you mentioned were addressed below. 

1. The search strategy seems well designed, and I appreciated the effort to analyze the quality of included studies and the possibility of publication bias. However, at least one recent full-study about CNNs that met the inclusion criteria was not included in the analysis (Urban 2018, Gastroenterology); also some abstracts (such as Misawa 2019 GIE, Matsui 2019 GIE, LUI 2019 GIE) were not included. The choice whether to include or not abstracts can be discussed (even if in a meta-analysis they are usually included), however abstracts were not included in the exclusion criteria.

Response: 

Urban 2018, Gastroenterology was already included in my analysis. I marked it in red in Table 1 and Table 2. It is also the number 30 in my reference list (in red). 

We have carefully read these abstracts (Misawa 2019 GIE, Matsui 2019 GIE, LUI 2019 GIE) and we found that they could not provide precise data for analysis. As a result, we decided not to include them. For more accurate expression, we added “abstracts” to the exclusion criteria part and marked it in red (Page 6, Line 132). 

2. Data extraction from the selected studies was not so rigorous and clear.

For example, apparently for reference 13 (Renner et al) authors have included the performances of CNN considering either high-confidence predictions or "standard" predictions (see table 2), with potential duplication of data and overestimation of performance.

Response: 

Thank you for this constructive suggestion. We carefully reviewed the article of Renner et al., and found maybe the high-confidence predictions had the potential of duplication of data. As a result, we decided to delete the data of high-confidence, and recalculated the whole part of classification. There are so many changes, and all of them were marked in red. (Page 16, Line 288, 289, 291, 292, 294, 295; Page 17, Line 307-309; Page 19, Line 346). Table 4, Table 6, Fig. 5, Fig. 6, Fig. 8, and Fig. 9 were remade. Besides, we added some notes to Fig. 3 (Page 15, Line 261-264), and Fig. 5 (Page 17, Line 298-299). All these changes were marked in red. 

3. Moreover, for reference 25 (Guo et al) the authors considered per-frame sensitivity and specificity of short and long videos; however, in order to provide clinically useful information (similar to polyp or adenoma detection rates) per-video sensitivity and specificity (reported in table 3 of the cited paper for the 100 short videos) should have been used instead. In table 3 of the paper by Guo et al are also reported the diagnostic performances of 2 experts e 2 non-experts physicians that were not included in the metanalysis.

Response: 

There are two reasons that we did use per-frame sensitivity and specificity instead of per-video sensitivity and specificity. First, almost of the articles included in our analysis used colonoscopy images instead of videos to compare the diagnostic performance between the CNN system and human endoscopists. In order to ensure the consistency of the whole analysis, we chose the per-frame sensitivity and specificity. Second, for per-video sensitivity and specificity, Guo et al. did not provide enough data for us to analyze. In table 3 of their paper, they only provide the data of per-video sensitivity, but not the data of per-video specificity. As a result, we cannot get the data of (TP, TN, FP, and FN). 

 When we read the paper of Guo et al., we have found that they reported the diagnostic performance of experts and non-experts. But the data of human endoscopists are per-video, not per-frame. Based on the reasons we mentioned above, we decided to use the data of per-frame instead of per-video. So we did not include the data of human endoscopists. 

4. Similarly, for Wang et al (ref 31) the 1633 polyps (and not the total number of polyps images) should have been considered.

Response:

The total 1633 polyps belong to Dataset A and C. 

In the paper of Wang et al., they used four databases to validate the diagnostic performance of CNN system. Dataset A and B contained colonoscopy images. Dataset C and D contained colonoscopy video. However, they only provide precise data of Dataset A. As a result, we chose included the data of Dataset A. When we search for the references, we have found almost all authors used colonoscopy images to compare diagnostic performance between CNN system and human endoscopists. So we decided to choose colonoscopy images as the main source of our analysis. 

5. I think that the use of the Fagan Nomogram to calculate post-test probability considering positive and negative likelihood ratios is not really indicated in this field.

Response: 

The Fangan Nomogram is based on Bayes’ Theorem. It is often used to evaluate the clinical application of a diagnostic method or tool. The pre-test probability is calculated using Bayes’ Theorem. The post-test probability of LR-positive means the probability of patients suspected of this disease would get a positive diagnostic result using this diagnostic method or tool. While the post-test probability of LR-negative means the probability of patients who are not suspected of this disease would suffer it after this diagnostic method or tool gives them a negative diagnostic result. 

After carefully thinking about the meaning of Fangan Nomogram, we think the application of it in the field of detection and classification of colorectal polyps may be appropriate. 

6. The differential performance according to dimension of polyps (diminutive and non-diminutive polyps) should have been analysed

Response:

Sometimes colorectal polyps are hard to detect and classify, especially for the diminutive ones. So maybe the diagnostic performance would be different between diminutive polyps and non-diminutive polyps. After carefully reviewed all the references, we found that seven studies provided data about diminutive polyps (Chen et al., Renner et al., Mori et al., Ozawa et al., Shin et al., Byrne et al., and Kudo et al.). Among them, Kudo et al. focused on the detection of Polyps, and Chen et al., Mori et al., and Byrne et al. mainly focused on diminutive polyps. Because only one study provided data about diminutive polyps in the detection part, so we thought it was not necessary to perform a subgroup analysis. In the subgroup analysis of diminutive polyps in the classification part, the result showed the performance of the CNN and expert had just slightly changed. Probably, the reason is because only two studies (Renner et al., and Ozawa et al.) provided data about non-diminutive polyps. After careful consideration, we decided not to add the subgroup analysis to the manuscript, due to the slight change of the results and the consistency of the article. 

The subgroup analysis was shown below:

Object Sensitivity (95% CI) Specificity

(95% CI) PLR 

(95% CI) NLR 

(95% CI) DOR 

(95% CI) SROC 

(95% CI)

CNN 0.935 [ 0.915-0.947] 0.770 [0.597-0.883] 4.061 [2.155-7.652] 0.085 [0.060-0.120] 47.849 [ 18.638-122.842] 0.983 [ 0.951-0.997]

Expert 0.912 [0.827-0.957] 0.848 [0.756-0.910] 6.009 [3.783-9.544] 0.104 [0.054-0.200] 57.970 [ 30.407-110.518] 0.938 [0.923-0.953]

Non-expert 0.878 [0.792-0.932] 0.782 [0.687-0.854] 3.915 [2.673-5.731] 0.174 [0.112-0.267] 25.428 [13.169-49.096] 0.895 [0.873-0.924]

Diminutive polyps 

CNN 0.948 [0.925-0.965] 0.812 [0.673-0.900] 5.037 [2.741-9.258] 0.064 [0.040-0.101] 79.210 [29.347-213.791] 0.965 [0.941-0.972]

Expert 0.903 [0.808-0.954] 0.864 [0.777-0.921] 4.659 [3.099-8.818] 0.112 [0.057-0.221] 49.389 [30.057-117.346] 0.921 [0.901-0.965]

Non-expert 0.878 [0.792-0.932] 0.782 [0.687-0.854] 3.915 [2.673-5.731] 0.174 [0.112-0.267] 25.428 [13.169-49.096] 0.895 [0.873-0.924]

Note: The sample size of non-diminutive polyps is too small, so it cannot be analyzed. The data of non-expert in whole group are identical to those in diminutive polyps group, so the results are just the same. 

7. The paper requires a deep linguistic revision (grammatical mistakes, repetitions…), possibly by an English mother-tongue revisor.

Response: 

We have already sent this paper to language polishing. (All changes were marked in blue.)

Other observations:

8. Reference 10 does not refer to detection of colonic lesions, but of early esophageal squamous cell carcinoma/preneoplastic lesions in the esophagus. The sentence in the introduction related to this reference should be revised.

Response:

We are sorry for this mistake. We have revised this mistake and cited another reference. 

9. Both table 1 and 2 should contain the reference numbers of the included papers in order to make the tables easier to read. Moreover, I suggest to list the papers according to the year of publication.

Response:

We added the reference number to Table 1 and 2 and listed papers according to year of publication. Besides, I fixed two references’ names and marked them in red. (Kuo is altered to Kudo; Yu is altered to Lequan )

10. Table 1 should also include more details about the imaging modalities included in the different studies. For example, the use of dedicated endocytoscopes in the studies referred as 12 and 25 should be reported. Also information about the use of real-time analysis by the CNN systems, and the type of lesions included (only diminutive polyps or polyps of any size) in the different studies should be highlighted in the table.

Response:

We have added information of type of endocytoscopes, type of CNN system, real-time use of CNN system, type of lesions, and type of images into Table 1, and marked them in red. 

11. In the paper it is stated that 7 of the included studies focused on detection; however in table 1 only 6 papers have detection been reported as field focus; similarly, 5 studies should include experts human endoscopists but only four papers are reported in table 1 and 2.

Response: 

Actually, there are 7 studies focused on detection, they are Lequan et al., Wang et al., Urban et al., Zhang et al., Yamada et al., Kudo et al., and Guo et al.. In fact, we made a mistake. There are only 4 studies provided data about experts and 3 studies provided data about non-experts. We revised it and marked it in red. (Page 10, Line 216)

12. The paper by Yu (2016), included in table 1 and 2 is not reported in the references.

Response:

Yu (2016) is actually Lequan (2016), we just mistook his/her last name. We have revised it and marked it in red. 

Review 3#

1. There are some spelling mistakes:

- line 186, page 15: Results instead of resules

- line 211, page 17, table 2: proximal rectosigmoid instead of poximal rectosigmoid

- line 360, page 25: ordinary physicians instead of phyisicians

Response:

We have revised all the spelling mistakes, and already sent the manuscript to language polishing.

2. There are missing the technical explanations of the different CNN-systems you compared in your analysis: Why are some CNN-systems better for CP detection and others for CP classification? Is it justified to compare different CNN-systems that provide different features? Do they have different characteristics concerning deep learning?

Response:

For CP detection training, authors chose images contained CP and those did not contain CP. Their purpose is to train the CNN system to distinguish CP from normal colorectal mucosa. This task is also very difficult. Because flat, small and isochromatic polyps are associated with a high miss rate, even for experts. 

 While for CP classification training, authors used images contained hyperplastic or neoplastic CP. The authors used NBI or magnification images to train CNN system to classify different pathological type of CP. For hyperplastic CP, the treatment is just simple resection. While for neoplastic CP, it is not enough. Endoscopic Mucosal Resection (EMR), Endoscopic submucosal dissection (ESD) or even surgery is the optimal curative option. If the accuracy of classification function of CNN system is high enough, endoscopists can trust the diagnostic results provided by it and make a simultaneous therapeutic decision while colonoscopy. 

 Maybe the details of algorithms of detection and classification are slightly different. But rare authors provided information about these differences. However, they are all based on CNN basically. 

3. There is also missing the description of the different classification-systems of CP used by different authors in different studies.

Response:

We have added information of different type of CNN system used in different studies in Table 1.

---

## [Decision Letter · Decision Letter 1]

24 Nov 2020

PONE-D-20-22694R1

Comparison of diagnostic performance between convolutional neural networks and human endoscopists for diagnosis of colorectal polyp: a systematic review and meta-analysis

PLOS ONE

Dear Dr. Xu,

Thank you for submitting your manuscript to PLOS ONE. After careful consideration, we feel that it has merit but does not fully meet PLOS ONE’s publication criteria as it currently stands. Therefore, we invite you to submit a revised version of the manuscript that addresses the points raised during the review process.

We look forward to receiving your revised manuscript.

Kind regards,

Ping He, Ph.D.

Academic Editor

PLOS ONE

Additional Editor Comments (if provided):

Please further polish your manuscript according to the comments below.

Reviewers' comments:

Reviewer's Responses to Questions

**Comments to the Author**

1. If the authors have adequately addressed your comments raised in a previous round of review and you feel that this manuscript is now acceptable for publication, you may indicate that here to bypass the “Comments to the Author” section, enter your conflict of interest statement in the “Confidential to Editor” section, and submit your "Accept" recommendation.

Reviewer #1: All comments have been addressed

Reviewer #2: (No Response)

Reviewer #3: All comments have been addressed

Reviewer #4: All comments have been addressed

Reviewer #5: All comments have been addressed

Reviewer #6: All comments have been addressed

Reviewer #7: All comments have been addressed

Reviewer #8: All comments have been addressed

2. Is the manuscript technically sound, and do the data support the conclusions?

Reviewer #1: Yes

Reviewer #2: Partly

Reviewer #3: Yes

Reviewer #4: Yes

Reviewer #5: Yes

Reviewer #6: Yes

Reviewer #7: Yes

Reviewer #8: Yes

3. Has the statistical analysis been performed appropriately and rigorously? 

Reviewer #1: Yes

Reviewer #2: No

Reviewer #3: Yes

Reviewer #4: Yes

Reviewer #5: Yes

Reviewer #6: Yes

Reviewer #7: I Don't Know

Reviewer #8: Yes

4. Have the authors made all data underlying the findings in their manuscript fully available?

Reviewer #1: Yes

Reviewer #2: Yes

Reviewer #3: Yes

Reviewer #4: Yes

Reviewer #5: Yes

Reviewer #6: Yes

Reviewer #7: Yes

Reviewer #8: Yes

5. Is the manuscript presented in an intelligible fashion and written in standard English?

Reviewer #1: Yes

Reviewer #2: Yes

Reviewer #3: Yes

Reviewer #4: No

Reviewer #5: Yes

Reviewer #6: Yes

Reviewer #7: Yes

Reviewer #8: Yes

6. Review Comments to the Author

Reviewer #1: I read the authors response. The revised manuscript is appropriately corrected. I have no comment for the authors.

Reviewer #2: (No Response)

Reviewer #3: I thank the authors for having carefully read through the comments of all three reviewers and that you made the proper revisions. Your responses to the reviewer’s questions are fully integrated in your revised manuscript that can now be accepted for publication.

Reviewer #4: The article provide valuable information. The manuscript is well organized. I would like authors to confirm if they have taken the informed consent. Also authors should rectify the grammatical mistakes and language errors in every paragraph. Authors are advise also to incorporate more citations and to put citation newer than 2010. I recommend the manuscript for the publication with minor revision.

Reviewer #5: I happen to be a follow-on reviewer and have been able to review your submissions, original as well as revised addressing the concerns of previous worthy reviewers. I found that the revised manuscript is much better and to me most of the adjustments, corrections and modifications have already been addressed. I would suggest to give some more consideration to the write-up in standard format of English.

Reviewer #6: As technology evolves it is more present in our personal lives and professional lives. We need to embrace because it is the future and there is no doubt about this. I congratulate the authors on their work regarding the volume of research and also on the starting hypothesis. We call these advanced structures AI or artificial intelligence but in essence they are algorithms which take into consideration past information on only selected types of lesions but they do not make logical or X crossed decisions. These aspects need to be taken into account always when discussing this technology-the lack of flexibility.

A few mentions:

There are some english mystakes. I have marked some of them in the attached document. Please adress them.

Line 102-104

102 Moreover, findings from some studies showed that the CNN system could automatically

103 classify CP, which is significantly helpful for the therapeutic decision-making

104 process during colonoscopy.

The authors mentions the CNN could classify CP. Please elaborate in a few words if this classification is based on malignancy risk or only on size or by base of implantation as this information has important value to the reader.

Line 101-102.

It is a type of the most prevalent network architectures of deep learning (DL) methods based on artificial intelligence (AI) technology.

Please reformulate-this phrase is pretty difficult to understand by the reader. I understand the information is abstract but it needs to make sense to everyone.

Line 214 the authors mention all of the articles were published in the last 4 years. In the abstract the end of the time-lime of the search was April 2020. Please also provide a starting date of the search if it exists.

Other aspects from my point of view that the authors should mention in a few words is the problem of ethics. How much can a doctor base his decision on an algorithm and what are the legal implications if a decision is wrong and the CP classified as benign proves to be malignant.

Another aspect which I did not see addressed is the suboptimal colonic preparation for colonoscopy. In current practice we have all encountered it. How does the CNN address these issues. In the studies included in the analysis were all of the patients prepared ideally for the colonoscopy?

Reviewer #7: well written systemic review and meta-analysis by Dr. Xu on role of artificial intelligence (AI) in medical science.

Reviewer #8: Interesting systematic review and meta-analysis for diagnostic performance of convolutional neural networks for the detection and classification of colorectal polyps. The authors have done a thorough job of addressing all of the prior reviewers questions and concerns appropriately. Grammatical and spelling mistakes have also been addressed. I have not identified any other major changes that need to be made, technical or otherwise.

7. PLOS authors have the option to publish the peer review history of their article (what does this mean?). If published, this will include your full peer review and any attached files.

Reviewer #1: No

Reviewer #2: No

Reviewer #3: **Yes: **Andreas Adler, M.D.

Reviewer #4: No

Reviewer #5: No

Reviewer #6: No

Reviewer #7: **Yes: **Dr. Irshad Ahmad

Reviewer #8: **Yes: **Stas Amato

---

## [Author Response · Author response to Decision Letter 1]

8 Dec 2020

Dear Pro. Ping He:

 Thank you and the reviewers for your valuable suggestions. We have carefully read through the comments and made proper revisions. Our responses to the reviewer’s questions are listed below. We greatly appreciated your time and efforts to improve our manuscript for publication.

Sincerely,

Xuezhong Xu

Reviewer 2#

1. It may be reasonable not to include abstracts in a meta-analysis. Thank you for having clarified this choice.

Response: Thank you.

2. Thank you for this major change. I think that this choice, such as the others subsequently listed, should somehow be discussed and motivated in the Methods section of the study. 

3. And 4. Thank you for these comments. Please try to discuss these choices in the method section.

Response: we discussed the choices of data extraction in the method section/data extraction and quality assessment. (they are in blue font, Line 146-162, Page 7-8). 

3. Of course I am aware of the meaning of the Fagan Nomogram; my doubts refer to the clinical, real life application of the Bayes’ theorem in this field. In particular, I criticize the arbitrary choice of the 20% pre-test probability of a patient having a polyp or of a polyp being an adenoma chosen in the paper. 

I find not clear the sentence “the pretest probability was defined as the prevalence of the target condition” (line 175, statistical analysis): The pre-test probability of a patient having a polyp of course will change accordingly to the patient’s characteristics and to indication for colonoscopy; the pre-test probability of a polyp being an adenoma depends from the characteristics of the polyp and is even more difficult to estimate in terms of %. 

If you want to keep the Fagan nomogram in the paper, I suggest choosing more appropriate pre-test probabilities (for example, expected polyp or adenoma detection rates in screening population derived from previous studies). However, I am really skeptical about the use of Fagan nomogram especially in the field of polyps’ classification: what does it mean that a polyp has 20% pre-test probability of being an adenoma? I think that an endoscopist generally has a much clearer idea of the nature of a polyp.

Response: After careful consideration, we thought the Fagan nomogram might not be suitable for this analysis. We chose to delete the results of Fagan nomogram. The statistical method of Fagan nomogram was deleted. The results about Fagan nomogram were deleted. The Fig 4 and Fig 6 were deleted. The order of other pictures was rearranged.

4. Thank you for the effort done to perform this subgroup analysis. Maybe you could consider inserting this subgroup analysis in the paper as a supplementary data, briefly discussing the results in the main paper.

Response: we added the subgroup analysis about diminutive polyps as S3 Table. We also discussed it in the discussion part (blue font, Line 425-431, Page 24).

5. The paper has for sure improved. However, I still find that some sections could be slightly modified in order to avoid repetitions (for example, lines 40-41 and 43-44 in the abstract; lines 205-206 and 210 in the results) and ease reading (line 447 in the discussion section). Some minor spelling mistakes are still present: line 187, he SROC (instead of the SROC); line 248, heterpgeneity; line 395, applid;

Response: 

Line 40-41 and 43-44 were modified. (blue font, Line 39-43, Page 2)

Line 205-206 and 210 were modified. (blue font, Line 216, Page 10)

Line 447 was modified. (blue font, Line 464-465, Page 25)

Line 187 was modified. (blue font, Line 198, Page 9)

Line 248 was modified. (blue font, Line 254, Page 15)

Line 395 was modified. (blue font, Line 398, Page 22)

6. Thanks to the changes done, tables 1 and 2 are easier to read now and more informative. I only suggest to modify “type of endocytoscopes” in table 1 with “type of endoscopes”, since endocytoscopes were used only in few papers.

Response: “endocytoscopes” was modified into “endoscopes”.

7. For the study by Kudo et al (reference 26), I also think that considering both White light images and NBI images may lead to duplication of data, because they refer to the same polyps.

Response: we carefully reviewed the reference, we found it was inappropriate to delete any of them. However, it is different from Renner et al.. In Renner et al. the images of standard-confidence predictions might be the same as the images of high-confidence predictions. In Kudo et al. images in different models are different. As a result, we chose to include both of WLI and NBI images. We added discussion about this choice in the data extraction and quality assessment part. (blue font, Line 157-162, Page 8)

8. In the abstract it is stated that “the diagnostic performance of the CNN system was superior to that of the expert and non-expert” (lines 43-44). However, the meta-analysis did not find any statistically significant difference. Modify the sentence accordingly.

Response: the sentence was modified into “the diagnostic performance of the CNN system was superior to that of the expert and non-expert in the field of CP classification, although the differences were not statistically significant”. (blue font, Line 41-43, Page 2)

9. I suggest to split the paragraph “The comparison of diagnostic performance among CNN system, expert and non-expert” (page 18), by positioning the comparisons in the fields of detection and classification immediately after the relative paragraphes “diagnostic performance of expert and non-expert”.

Response: the paragraph was split and repositioned immediately after the relative paragraphs “diagnostic performance of expert and non-expert”. 

Reviewer 4#

1. The article provide valuable information. The manuscript is well organized. I would like authors to confirm if they have taken the informed consent. Also authors should rectify the grammatical mistakes and language errors in every paragraph. Authors are advise also to incorporate more citations and to put citation newer than 2010. I recommend the manuscript for the publication with minor revision.

Response: All the authors have taken the informed consent and have no conflict of interest to disclose. I am sorry for grammatical mistakes and language errors. We have re-polished the whole article. All the places that have been modified were marked in red font.

 We have checked the reference list, and we found that reference 4, 8, 15, 16, 18, 19, 20, and 21 were published before 2010. We changed reference 4 and 8 into newest ones. The rest of them are about the statistical methods of meta-analysis, which have been invented a long time ago. So we did not change the rest of the references. 

Reviewer #5: 

I happen to be a follow-on reviewer and have been able to review your submissions, original as well as revised addressing the concerns of previous worthy reviewers. I found that the revised manuscript is much better and to me most of the adjustments, corrections and modifications have already been addressed. I would suggest to give some more consideration to the write-up in standard format of English.

Response: We have re-polished the manuscript. We hope that it will be in the standard format of English. All the places that have been modified were marked in red font.

Reviewer #6: 

1. As technology evolves it is more present in our personal lives and professional lives. We need to embrace because it is the future and there is no doubt about this. I congratulate the authors on their work regarding the volume of research and also on the starting hypothesis. We call these advanced structures AI or artificial intelligence but in essence they are algorithms which take into consideration past information on only selected types of lesions but they do not make logical or X crossed decisions. These aspects need to be taken into account always when discussing this technology-the lack of flexibility.

Response: we have added the limitations of AI to the discussion part. (blue font, Line 451-457, Page 25)

2. There are some english mystakes. I have marked some of them in the attached document. Please adress them.

Response: we have corrected all the mistakes. 

Line 194 “informed consent form”: we deleted the ethics part, because the editor said it is not needed in the meta-analysis. 

Line 209 “rules out” into “excluded”

Line 367-368 ”despite the differences were statistically insignificant” into “although the differences were statistically insignificant”

Line 393 “qualify being an expert” into “However, it is vital to consider that not all endoscopists possess expert experience”

Line 398 “applid” into “harbor the application”

3. Line 102-104

102 Moreover, findings from some studies showed that the CNN system could automatically

103 classify CP, which is significantly helpful for the therapeutic decision-making

104 process during colonoscopy.

The authors mentions the CNN could classify CP. Please elaborate in a few words if this classification is based on malignancy risk or only on size or by base of implantation as this information has important value to the reader.

Response: The CNN system could classify CP based on its morphological features. we have added this information to this sentence. (red font, Line 104-105, Page 5)

4. Line 101-102.

It is a type of the most prevalent network architectures of deep learning (DL) methods based on artificial intelligence (AI) technology.

Please reformulate-this phrase is pretty difficult to understand by the reader. I understand the information is abstract but it needs to make sense to everyone.

Response: We found it was difficult to reformulate this sentence. In order to make everyone understand the concept about AI, DP and CNN, we added a figure (S1 Fig) to this abstract.

5. Line 214. the authors mention all of the articles were published in the last 4 years. In the abstract the end of the time-lime of the search was April 2020. Please also provide a starting date of the search if it exists.

Response: The application of CNN in the field of colonoscopy has been introduced recently. When we searched for the relevant articles, we did not set a starting date. The result showed all studies about CNN in colonoscopy were published in the last 4 years.

6. Other aspects from my point of view that the authors should mention in a few words is the problem of ethics. How much can a doctor base his decision on an algorithm and what are the legal implications if a decision is wrong and the CP classified as benign proves to be malignant.

Response: We added a few sentences in the discussion part to declare the problem of ethics. (red font, Line 432-435, Page 24)

7. Another aspect which I did not see addressed is the suboptimal colonic preparation for colonoscopy. In current practice we have all encountered it. How does the CNN address these issues. In the studies included in the analysis were all of the patients prepared ideally for the colonoscopy?

Response: All the article included in my analysis extracted images with high quality (high-definition and well bowel preparation). As a result, it will lead to selection bias. Meanwhile, the performance of CNN system may be excellent in the training set but weak in the clinical practice. (Line 436-439, Page 24)

 Besides, there is also an AI system that specially evaluates bowel preparation [1]. 

1. Zhou J, Wu L, Wan X, Shen L, Liu J, Zhang J, et al. A novel artificial intelligence system for the assessment of bowel preparation (with video). Gastrointest Endosc. 2020;91(2):428-35.e2. Epub 2019/11/30. doi: 10.1016/j.gie.2019.11.026. PubMed PMID: 31783029.

---

## [Decision Letter · Decision Letter 2]

26 Jan 2021

PONE-D-20-22694R2

Comparison of diagnostic performance between convolutional neural networks and human endoscopists for diagnosis of colorectal polyp: a systematic review and meta-analysis

PLOS ONE

Dear Dr. Xu,

Thank you for submitting your manuscript to PLOS ONE. After careful consideration, we feel that it has merit but does not fully meet PLOS ONE’s publication criteria as it currently stands. Therefore, we invite you to submit a revised version of the manuscript that addresses the points raised during the review process.

We look forward to receiving your revised manuscript.

Kind regards,

Ping He, Ph.D.

Academic Editor

PLOS ONE

Additional Editor Comments (if provided):

This manuscript can be accepted with additional conditions. Ask the author to contribute to one of the reviewers' questions. The author is requested to complete the revision and send it to me for review and approval.

Reviewers' comments:

Reviewer's Responses to Questions

**Comments to the Author**

1. If the authors have adequately addressed your comments raised in a previous round of review and you feel that this manuscript is now acceptable for publication, you may indicate that here to bypass the “Comments to the Author” section, enter your conflict of interest statement in the “Confidential to Editor” section, and submit your "Accept" recommendation.

Reviewer #1: All comments have been addressed

Reviewer #2: All comments have been addressed

Reviewer #3: All comments have been addressed

Reviewer #5: All comments have been addressed

Reviewer #6: All comments have been addressed

Reviewer #7: All comments have been addressed

2. Is the manuscript technically sound, and do the data support the conclusions?

Reviewer #1: Yes

Reviewer #2: Partly

Reviewer #3: Yes

Reviewer #5: Yes

Reviewer #6: Yes

Reviewer #7: Yes

3. Has the statistical analysis been performed appropriately and rigorously? 

Reviewer #1: Yes

Reviewer #2: No

Reviewer #3: Yes

Reviewer #5: Yes

Reviewer #6: Yes

Reviewer #7: Yes

4. Have the authors made all data underlying the findings in their manuscript fully available?

Reviewer #1: Yes

Reviewer #2: Yes

Reviewer #3: Yes

Reviewer #5: Yes

Reviewer #6: Yes

Reviewer #7: Yes

5. Is the manuscript presented in an intelligible fashion and written in standard English?

Reviewer #1: Yes

Reviewer #2: Yes

Reviewer #3: Yes

Reviewer #5: Yes

Reviewer #6: Yes

Reviewer #7: Yes

6. Review Comments to the Author

Reviewer #1: The revised manuscript has bee revised appropriately. I have no comment for the authors.

Reviewer #2: I would like to thank the authors for the efforts done in order to answer to all the comments provided. The revisions increased the general quality of the paper, however I still have some doubts about methodological/statistical aspects of the meta-analysis that have to be addressed.

The most important aspect of a meta-analysis is the reproducibility of the analysis by external readers.

I carefully analyzed the results showed in table 2. In this table are reported the diagnostic categories (TP, FP, TN, FN) of all the studies included in the meta-analysis. I tried to calculate by myself sensitivity for detection and I found different values from the ones reported in the paper. For example, considering only the seven papers labeled as “detection” studies in table 1, I calculated for the CNN system a sensitivity (that is TP/TP+FN) of 64902/77412=0.838, that is different from the reported 0.909.

After careful review, I have noticed that the study by Kudo et al [17] has been erroneously classified as detection study in table 1, while it is a classification study. Even excluding this study, however, sensitivity for CNN for detection is different (60462/77252= 0.782) from the reported one.

I have also another doubt: were all the data included in table 2 used to calculate diagnostic performances? If yes, there was a duplication of small lesions in Kudo [17] (reported both all the lesions, that include small lesions, and separately only lesions < 5 mm), Ozawa [28] (all lesions and only lesions < 10 mm) and Renner [15] (all lesions and diminutive rectosigmoid).

1. I suggest to split table 2 in two different tables, one including only detection studies and one classification studies. This will make easier for readers (and for reviewers) to independently control the diagnostic performances reported in the paper and will allow to clarify the reasons for the difference I found in detection sensitivity.

2. Modify the reported “field” of the study by Kudo [17] in table 1 (detection � classification). I suggest to carefully review all the papers included in order to correctly classify them into detection and classification studies.

3. I suggest to carefully review again all the data inserted (for example, I noticed that for Renner et al. are reported 99 results in table 2, but the original paper includes 100 polyps) and to recalculate all the reported diagnostic performances

4. Were small lesions in Kudo, Ozawa and Renner considered twice as they were reported in table 2? If yes, this is a mistake that has to be corrected (duplication of small polyps)

One more limitation of the study is that more than half of the data come from a single study (Guo et al [16], especially because of the inclusion of “full-videos” data. The results for detection are, for this reason, strongly “guided” by this study.

5. Can you provide, in addition to the general analysis, another analysis for detection excluding the large study by Guo et al? It could be very informative and add strength to the meta-analysis results. You should also briefly discuss this aspect in the discussion

Below are listed my comments to the revisions made in response to my previous comments:

2. Response: we discussed the choices of data extraction in the method section/data extraction and quality assessment. (they are in blue font, Line 146-162, Page 7-8).

Thank you for the explanations provided.

3. Response: After careful consideration, we thought the Fagan nomogram might not be suitable for this analysis. We chose to delete the results of Fagan nomogram. The statistical method of Fagan nomogram was deleted. The results about Fagan nomogram were deleted. The Fig 4 and Fig 6 were deleted. The order of other pictures was rearranged.

Thank you for having taken in consideration my observations.

4. Response: we added the subgroup analysis about diminutive polyps as S3 Table. We also discussed it in the discussion part (blue font, Line 425-431, Page 24).

Thank you. I suggest to briefly describe main results of this subgroup analysis in the results section.

7. Response: we carefully reviewed the reference, we found it was inappropriate to delete any of them. However, it is different from Renner et al. In Renner et al. the images of standard-confidence predictions might be the same as the images of high-confidence predictions. In Kudo et al. images in different models are different. As a result, we chose to include both of WLI and NBI images. We added discussion about this choice in the data extraction and quality assessment part. (blue font, Line 157-162, Page 8)

Thank you for having clarified the differences between the two papers. Even if I am still convinced that considering both the modalities may include a risk of duplication of data (and, more in general, that the use of “per frame” analysis instead of “per lesion” analysis may be misleading and not applicable to clinical practice), now I can understand this choice.

I suggest including in the discussion a few sentences about the possibly reduced translational applicability of the results of this metanalysis because of the use of per frames and per video data: in clinical practice it is important to identify and classify a specific polyp, not all the images regarding the polyp itself.

8. Response: the sentence was modified into “the diagnostic performance of the CNN system was superior to that of the expert and non-expert in the field of CP classification, although the differences were not statistically significant”. (blue font, Line 41-43, Page 2)

Thank you for the change. However, there is a spelling mistake in line 43: were not statistically insignificant instead of were not statistically significant.

9. Response: the paragraph was split and repositioned immediately after the relative paragraphs “diagnostic performance of expert and non-expert”.

Thank you.

Minor concerns:

- Lines 27-28: I find too severe the sentence “significantly unsatisfactory” to describe the efficacy of colonoscopy.

- Line 71: I don’t find the word approaches adequate in this context. I suggest to modify it in “mechanisms”

- Line 106-107: The new sentence “Nevertheless, this technology has not reached maturity hence unsatisfactory.” seems incomplete to me.

- I suggest to include some statistical results (at least p-values) in the abstract.

Reviewer #3: (No Response)

Reviewer #5: I believe it is a much better and appropriate manuscript after the second revision. I congratulate the researchers/authors for doing a wonderful job.

Reviewer #6: The authors have addressed the raised issues with the article. From my personal point of view the paper is fit for publication.

Reviewer #7: The present revision of the manuscript is well polished and much better than the original submission of the same. The reviewer comments have also been responded well.

7. PLOS authors have the option to publish the peer review history of their article (what does this mean?). If published, this will include your full peer review and any attached files.

Reviewer #1: No

Reviewer #2: No

Reviewer #3: **Yes: **Andreas Adler, M.D.

Reviewer #5: **Yes: **M Amir

Reviewer #6: No

Reviewer #7: No

---

## [Author Response · Author response to Decision Letter 2]

26 Jan 2021

Dear Pro. Ping He:

 Thank you and the reviewers for all your valuable suggestions. We have carefully read through the comments and made proper revisions. Our responses to the reviewer’s questions are listed below. We greatly appreciated your time and efforts to improve our manuscript for publication.

Sincerely,

Xuezhong Xu

Reviewer 2#

1. I carefully analyzed the results showed in table 2. In this table are reported the diagnostic categories (TP, FP, TN, FN) of all the studies included in the meta-analysis. I tried to calculate by myself sensitivity for detection and I found different values from the ones reported in the paper. For example, considering only the seven papers labeled as “detection” studies in table 1, I calculated for the CNN system a sensitivity (that is TP/TP+FN) of 64902/77412=0.838, that is different from the reported 0.909.

Response: We recalculated the whole analysis. We do the analysis using the midas package of STATA. We found the result of pooled sensitivity or specificity was not simply calculated as TP/(TP+FN). The data must be transformed somehow according to the heterogeneity. As a result, maybe the sensitivity or specificity you calculated was not right.

2. After careful review, I have noticed that the study by Kudo et al [17] has been erroneously classified as detection study in table 1, while it is a classification study. Even excluding this study, however, sensitivity for CNN for detection is different (60462/77252= 0.782) from the reported one.

Response: We modified Kudo to the field of CP classification and do all the analysis again.

3. I have also another doubt: were all the data included in table 2 used to calculate diagnostic performances? If yes, there was a duplication of small lesions in Kudo [17] (reported both all the lesions, that include small lesions, and separately only lesions < 5 mm), Ozawa [28] (all lesions and only lesions < 10 mm) and Renner [15] (all lesions and diminutive rectosigmoid).

Response: We deleted all the data which had the potential risk of duplication. Initially, we wanted to perform a subgroup analysis for diminutive polyps, but STATA could not analyze data with sample size smaller than 4.

4. I suggest to split table 2 in two different tables, one including only detection studies and one classification studies. This will make easier for readers (and for reviewers) to independently control the diagnostic performances reported in the paper and will allow to clarify the reasons for the difference I found in detection sensitivity.

Response: We split Table 2 into two different tables.

5. Modify the reported “field” of the study by Kudo [17] in table 1 (detection � classification). I suggest to carefully review all the papers included in order to correctly classify them into detection and classification studies.

Response: We modified Kudo to the field of CP classification and do all the analysis again.

6. I suggest to carefully review again all the data inserted (for example, I noticed that for Renner et al. are reported 99 results in table 2, but the original paper includes 100 polyps) and to recalculate all the reported diagnostic performances.

Response: We checked all data included again and modified the data of Renner et al.

7. Were small lesions in Kudo, Ozawa and Renner considered twice as they were reported in table 2? If yes, this is a mistake that has to be corrected (duplication of small polyps)

Response: We deleted all the data which had the potential risk of duplication. Initially, we wanted to perform a subgroup analysis for diminutive polyps, but STATA could not analyze data with sample size smaller than 4.

8. One more limitation of the study is that more than half of the data come from a single study (Guo et al [16], especially because of the inclusion of “full-videos” data. The results for detection are, for this reason, strongly “guided” by this study.

Can you provide, in addition to the general analysis, another analysis for detection excluding the large study by Guo et al? It could be very informative and add strength to the meta-analysis results. You should also briefly discuss this aspect in the discussion.

Response: We did subgroup analysis without the data of Guo et al. and discussed it in Line 392-396, Page 20.

9. Minor concerns:

Minor concerns:

- Lines 27-28: I find too severe the sentence “significantly unsatisfactory” to describe the efficacy of colonoscopy.

- Line 71: I don’t find the word approaches adequate in this context. I suggest to modify it in “mechanisms”

- Line 106-107: The new sentence “Nevertheless, this technology has not reached maturity hence unsatisfactory.” seems incomplete to me.

- I suggest to include some statistical results (at least p-values) in the abstract.

Response: We modified all the minor concerns.

---

## [Editor Report · Decision Letter 3]

28 Jan 2021

Comparison of diagnostic performance between convolutional neural networks and human endoscopists for diagnosis of colorectal polyp: a systematic review and meta-analysis

PONE-D-20-22694R3

Dear Dr. Xu,

We’re pleased to inform you that your manuscript has been judged scientifically suitable for publication and will be formally accepted for publication once it meets all outstanding technical requirements.

Kind regards,

Ping He, Ph.D.

Academic Editor

PLOS ONE

Additional Editor Comments (optional):

The authors have addressed all the questions issued by the reviewers. I think the quality of the paper has been greatly improved. I suggest publishing it directly.
---

## [Editor Report · Acceptance letter]

4 Feb 2021

PONE-D-20-22694R3 

Comparison of diagnostic performance between convolutional neural networks and human endoscopists for diagnosis of colorectal polyp: a systematic review and meta-analysis 

Dear Dr. Xu:

I'm pleased to inform you that your manuscript has been deemed suitable for publication in PLOS ONE. Congratulations! Your manuscript is now with our production department. 

Kind regards, 

on behalf of

Prof. Ping He 

Academic Editor

PLOS ONE